# Changes in the Rhizosphere Prokaryotic Community Structure of *Halodule wrightii* Monospecific Stands Associated to Submarine Groundwater Discharges in a Karstic Costal Area

**DOI:** 10.3390/microorganisms11020494

**Published:** 2023-02-16

**Authors:** Alonso de la Garza Varela, M. Leopoldina Aguirre-Macedo, José Q. García-Maldonado

**Affiliations:** Centro de Investigación y de Estudios Avanzados del Instituto Politécnico Nacional, Mérida 97310, Yucatán, Mexico

**Keywords:** seagrass, submerged groundwater discharges, rhizosphere, 16S rRNA

## Abstract

Belowground seagrass associated microbial communities regulate biogeochemical dynamics in the surrounding sediments and influence seagrass physiology and health. However, little is known about the impact of environmental stressors upon interactions between seagrasses and their prokaryotic community in coastal ecosystems. Submerged groundwater discharges (SGD) at Dzilam de Bravo, Yucatán, Mexico, causes lower temperatures and salinities with higher nutrient loads in seawater, resulting in *Halodule wrightii* monospecific stands. In this study, the rhizospheric archaeal and bacterial communities were characterized by 16S rRNA Illumina sequencing along with physicochemical determinations of water, porewater and sediment in a 400 m northwise transect from SGD occurring at 300 m away from coastline. Core bacterial community included *Deltaproteobacteria*, *Bacteroidia* and *Planctomycetia*, possibly involved in sulfur metabolism and organic matter degradation while highly versatile *Bathyarchaeia* was the most abundantly represented class within the archaeal core community. Beta diversity analyses revealed two significantly different clusters as result of the environmental conditions caused by SGD. Sites near to SGD presented sediments with higher redox potentials and sand contents as well as lower organic matter contents and porewater ammonium concentrations compared with the furthest sites. Functional profiling suggested that denitrification, aerobic chemoheterotrophy and environmental adaptation processes could be better represented in these sites, while sulfur metabolism and genetic information processing related profiles could be related to SGD uninfluenced sites. This study showed that the rhizospheric prokaryotic community structure of *H. wrightii* and their predicted functions are shaped by environmental stressors associated with the SGD. Moreover, insights into the archaeal community composition in seagrasses rhizosphere are presented.

## 1. Introduction

Seagrass meadows perform a variety of ecosystem services including carbon sequestration, nutrient cycling, water clarity enhancement and primary food sources [1,2]. As the only marine angiosperms, seagrasses root system oxygenates the surrounding thin zone of sediment, known as rhizosphere [3]. In this zone, oxygen and dissolved organic carbon, such as sucrose, recently found as the major sugar excreted by seagrasses, leak from roots creating a dynamic environment for a diverse group of microorganisms critical for plant nutrient acquisition, host defense to pathogens and biogeochemical cycling [3,4,5,6,7]. Dominant bacterial members in belowground compartments usually belong to the classes *Alpha-*, *Gamma-*, *Delta-*, *Epsilonproteobacteria* and *Bacteroidetes* [3], and the three most important functional groups for seagrass ecology are considered aerobic heterotrophs, sulfate-reducing and nitrogen fixing bacteria [5]. Since there is a variety of reports on historical events of seagrasses die-off [8] caused by eutrophication and the accumulation of the phytotoxin H_2_S, denitrification [1,9,10] and sulfide-oxidation these processes [5,8] are also important for seagrass health and survival. Although archaeal groups account for a significant part of the prokaryotic community in coastal ecosystems, little attention has been paid to seagrasses colonized sediments [11]. Current research has found *Euryarchaeota*, *Bathyarchaeota*, *Crenarchaeota*, *Thermoplasmatota* and *Asgardarchaeota* as the most abundant taxa in *Zostera Japonica* and *Zostera noltii* meadows [12,13]. Some archaeal taxa, such as *Bathyarchaeota*, may be involved in important biogeochemical functions such as sedimentary organic matter degradation, acetogenesis and methane metabolism [11,14,15]. Thereby, information on archaeal community structure and related functions could improve the understanding of the functioning in this type of ecosystems.

Despite the plethora of ecosystem services provided by seagrass meadows, growing environmental pressures associated with nutrient enrichment and light reduction are considered mayor drivers for worldwide seagrass losses at an unprecedented rate, estimated at 2–5% per year [16,17,18,19]. For instance, freshwater discharges may increase high-nutrient loads and negatively affect photosynthesis and seagrass growth by decreasing seawater salinity [20,21,22]. Increased nitrogen concentrations in the water column may increase amine exudation by roots and a decrease in non-structural carbohydrates reserves and growth rates [18,23]. Although below-ground interactions are governed by photosynthetic rates and root exudates, and microbial communities respond rapidly to environmental disturbance, making them suitable to monitor seagrass ecological status, most descriptors these days on seagrass monitoring programs rely on seagrass population responses, such as species composition, cover percentage, density, etc. [5,16,24]. Moreover, with the exception of few studies describing natural environmental gradients [4,25], scarce information of specific environmental stressors (particularly in sediments and porewater) and their influence over belowground associated microbial communities, is available [18,19]. Microbial research on seagrass rhizospheres have mainly focused in *Zostera marina* [3,26], *Cymodocea nodosa* [3], *Thallassia testudinum* and *Syringodium filiforme* [27] while other seagrass species have received little attention. Such is the case of *Halodule wrightii*, widely distributed in every continent, except Antarctica, considered an important primary producer and an early colonizer after environmental perturbations, preceding climax tropical species such as *T. testudinum* and *S. filiforme* [28,29]. Given its broader tolerance to environmental conditions compared to local seagrasses [29,30], this species could be considered a desirable candidate for assessing microbial interactions under the effect of environmental stressors.

Whether seagrass species harbor their own unique microbiome or it is shaped by environmental conditions, remain still to be addressed [7,16]. These aspects are among the main future issues to monitor seagrass ecological status, especially in eutrophic coastal zones [16]. Therefore, this study aimed (i) to identify the core microbiome of *H. wrightii* monospecific stands rhizospheres along a natural environmental gradient of 400 m generated by SGD in a karstic coastal area, located 300 m from coastline, (ii) to assess if prokaryotic community structure differed across sampling sites and to determine the environmental drivers causing such changes, and (iii) to infer potential roles of prokaryotic taxa using functional profiling across sites.

## 2. Materials and Methods

### 2.1. Site Description and Sampling

Dzilam de Bravo locates in the central region of Yucatán northern coast area (Figure 1a) and presents a semi-arid climate with temperatures ranging from 24 to 36 °C [30]. As other regions in Yucatán Peninsula, continental water flows from south to north through pores, fissures and fractures [31]. In this region SGD distribute in the area mixing seawater and groundwater, thus producing estuarine conditions in the sea [32]. X’Buya-Ha spring is located approximately 350 m from the coastline (Figure 1b) and is the most energetic discharge of the Yucatán coast, with water flow velocities up to 2 m s^−1^ depending on the tide, nutrient inputs and reductions in background salinity (36) to values ranging from 21 to 23 [30,33,34]. Despite other seagrass species develop in the area, only *H. wrightii* monospecific stands can be found near X’Buya-Ha spring with *S. filiforme* and *T. testudinum* dominance increasing with distance from the spring [30]. This being explained by the higher nutritional requirements of *H.wrightii* and its broader tolerance to environmental stressors [29].

Rhizospheres from *H. wrightii* monospecific stands, water, surrounding sediment and porewater samples were collected in August 2021 along a 400 m transect under the influence of submarine groundwater discharges (SGD) in Dzilam de Bravo, Yucatán, México (Figure 1b). Six sampling stations (S1–S6) were established starting at 32 m north of X’Buya-Ha spring (Figure 1b). Less energetic springs were also observed near stations S3 and S4. As reported before [30], *H.wrightii* was the only seagrass species that could be observed in sites S1–S4, while *S. filiforme* and *T. testudinum* patches were only observed in sites S5 and S6. Distances from X’Buya-Ha spring and geographic locations for sampling sites are shown in Appendix A.

Three 15 cm diameter sediment cores were randomly extracted in each sampling station from a depth of 10–15 cm. pH/redox values were obtained with an HI 2213 instrument (Hanna Instruments). Seagrass’ roots were manually shaken to remove loose sediment and collected with only the attached sediment (rhizosphere) in cryovials, and flash frozen in liquid nitrogen. The remaining sediments were collected separately for further physicochemical analysis. Rhizospheres were obtained by detaching the sediment fraction from the seagrass’s roots using a multi-step phosphate buffer washing and centrifugation procedure [35,36]. Porewater samples were obtained by using a custom-made device attached to a sterile syringe-hose system and, then, filtered through 0.45 µm syringe filters. Water samples were collected 10 cm above seagrass canopy using sterile plastic bottles. Salinity and temperature measurements were recorded in situ using a YSI 556 multiparameter system. All liquid samples were stored at 4 °C in plastic bottles until further analysis.

### 2.2. Environmental Characterization

Total nitrogen (NH_4_^+^ + NO_2_^−^ + NO_3_^−^) and soluble reactive phosphorus (PO_4_^3−^) in water as well as ammonium (NH_4_^+^) for porewater samples were determined according to [37]. Particle size determination and organic matter content (O.M%) in sediments were determined according to [38] and [39], respectively. Total carbon (TC%) and Total Nitrogen (TN%) in sediments were determined via gas chromatography with an elemental autoanalyzer (Flash EA-1112). Total phosphorus (TP%) was extracted from sediments and quantified as reported by [40].

### 2.3. DNA Extraction and 16S rRNA Gene Sequencing

DNA was extracted with the commercial kit DNeasy PowerSoil Kit (Qiagen, Hilden, Germany) using 0.5 g of rhizosphere samples (fresh weight) and following manufacturer’s instructions. DNA quality was verified by agarose gel 1%. Prokaryotic (archaea and bacteria) community characterization was analyzed using the primers 515 F-Y (5′GTGYCAGCMGCCGCGGTAA-3′) and 926R (5′-CCGYCAATTYMTTTRAGTTT-3′) covering the V4 and V5 hyper-variable regions of the 16S rRNA gene [41]. Polymerase chain reactions were performed for each triplicate of rhizosphere samples for all sites (S1-S6) and visualized on a 2% agarose gel. Reactions were performed in 20 µL final volumes containing 10 µL of Phusion Flash High-Fidelity Master Mix (Thermo Scientific, Waltham, MA, USA), 0.5 µL of each primer, 7 µL of PCR grade water and 2 µL of extracted DNA. PCR conditions involved an initial denaturation at 95 °C for 2 min, followed by 28 cycles of 45 s at 95 °C, 45 s at 52 °C, 90 s at 68 °C. The final elongation step was conducted at 68 °C for 5 min. PCR products were indexed using Nextera XT Index Kit v2 (Illumina, San Diego, CA, USA), gene amplicon libraries prepared according to llumina’s 16S Metagenomic Sequencing Library Preparation protocol and sequenced with an Illumina MiSeq instrument at CINVESTAV Mérida in a 2 × 250-bp paired-end run). The datasets generated in this study can be found in the online repository of the National Center for Biotechnology Information. The data are available under the BioProject number PRJNA927344.

### 2.4. Bioinformatic Analysis

Demultiplexed sequences were imported into the Quantitative Insights Into Microbial Ecology (QIIME 2) pipeline. Quality filter, trimming and denoising was conducted using the Divisive Amplicon Denoising Algorithm 2 (DADA2) plugin, with the “consensus” method for chimeras removal. Amplicon Sequence Variants (ASV) assignment was conducted using SILVA 132 16S rRNA gene database. The R environment was used for the removal of Chloroplast, Mitochondria and Unassigned sequences with packages phyloseq and MetagMisc [42,43]. Phyloseq tax table problematic entries were fixed with MicroViz [44]. A subsequent rarefaction step was conducted to the lowest sequencing depth for all samples and assessed with rarefaction curves using the MicrobiotaProcess package [45].

All metrics, plots and statistical analysis for alpha diversity, relative abundances, beta diversity, differential abundance analysis and explainable factors (environmental data) analysis relied on microeco and ggplot2 packages [46,47]. Kruskal-Wallis tests followed by Dunn’s post tests were conducted on environmental data and alpha diversity metrics to determine differences between sites. Six-way petal plots were generated for core community detection based on Venn analysis [48]. PERMANOVA tests were carried out to determine differences for beta-diversity (Bray-Curtis distances). Differential abundances analysis using LEFSE (α = 0.05) was conducted for determining significant taxa and community differences across rhizosphere sites [49].

For evaluating significant environmental factors influence over taxa, Spearman’s correlation heatmaps using fdr correction were generated. Additionally, distance-based redundancy analysis (db-RDA) using Bray-Curtis distances and Mantel tests (Spearman) were used to determine drivers of community structure.

Functional profiling for prokaryotic communities of rhizosphere samples was predicted with the subsystem level 3 of KEGG Orthologues using Tax4Fun [50] within the microeco package. A LEFSE analysis (α = 0.05) with fdr correction was conducted to find the differences across sites. Prokaryotic clades were mapped against the FAPROTAX database using the microeco package to predict relevant ecological functions. Taxonomic community profiles were converted into putative functional profiles and functional individual percentages were calculated considering the abundance of taxa [46,51].

## 3. Results

### 3.1. Environmental Characteristics

An environmental gradient of decreasing salinity and temperature but increasing orthophosphate and inorganic nitrogen concentrations in water samples resulted from the influence of X’Buya-Ha spring. Increasing sand content and redox potentials, but decreasing organic matter content in sediment samples, were also detected near the SGD. Lower ammonium concentrations (except site S1) and pH values in porewater samples were obtained for all the sites. Environmental characteristics for water, porewater and sediments are presented in Table 1 and Table 2.

Kruskal-Wallis followed by Dunn’s post-test analysis (Appendix A) showed that significant differences were observed between those sites closer to the spring (S1 and S2) and the furthest site (S6). All measured parameters for water samples were significantly different (*p* < 0.01) among sampling sites. Organic matter, sand content and total nitrogen were different (*p* < 0.05) for sediment characteristics. Ammonium concentration was the only different factor (*p* < 0.01) for porewater samples. Spearman’s correlations among environmental factors shows organic matter and porewater (PW) ammonium were positively correlated (r = 0.644, *p* < 0.01) with each other. PW ammonium (r = −0.635, *p* < 0.01) and organic matter (r = −0.542, *p* < 0.05) negatively correlated with sand and positively correlated with water salinity. A complete correlations matrix and complementary plots is shown in Appendix A.

### 3.2. Summary of 16S rRNA Data, Microbial Diversity, and Microbial Community Composition

A total of 411,484 raw reads were obtained for all the samples described in this study. 236,473 reads and 4737 amplicon sequence variants (ASVs) resulted after the quality filtering. Rarefaction curves obtained by the normalization step showed that the number of ASV reached a plateau suggesting an adequate sampling effort (Appendix A). In this case 93% of all ASVs accounted for bacteria and 7% for archaea. Furthermore, 54 phyla, 104 classes, 162 orders, 179 families and 190 genera were grouped.

Alpha-diversity metrics are shown in Figure 2. Shannon index (panel a) ranged from 5.0 ± 0.3 to 5.7 ± 0.2 and Simpson index (panel b) ranged from 0.097 ± 0.012 to 0.099 ± 0.002. No significant differences were found across sites, as resulted by Dunn’s Kruskal-Wallis analysis.

Most abundant bacterial phyla in rhizosphere samples were *Proteobacteria* (21.8–37.8%), *Bacteroidetes* (21–32.8%) and *Acidobacteria* (6.2–12.4%) (Figure 3a), while *Bacteroidia* (15.8–28.1%), *Deltaproteobacteria* (10.2–21.5%) and *Gammaproteobacteria* (4.5–23.3%) were the classes better represented (Figure 3b). Archaeal top phyla were *Crenarchaeota* (1.6–6.2% of prokaryotic community), *Euryarchaeota* (0.1–3.2%) and *Asgardeota* (0.8–3.3%) and top classes included *Bathyarchaeia* (1.6–6.2%), *Lokiarchaea* (0.8–3.3%) and *Thermoplasmata* (0.1–2.9%) (Figure 3). 96% of archaeal community accounted for *Crenarchaeota* (47.4% average relative abundance), *Euryarchaeota* (24.4%) and *Asgardaeota* (24.4%). *Bathyarchaeia* class accounted for 47% of archaeal sequences.

Core microbial community at class, family and genus taxonomic levels was obtained by the overlapping of all rhizosphere samples in a petal diagram and represent 5.5% (178) of total ASVs (Appendix A). Most abundant taxa included *Bathyarchaeia* and *Lokiarchaeia* classes. *Desulfobulbaceae*, *PHOS-HE36*, *Bacteroidetes BD-2*, *Calditrichaceae*, *Sandaracinaceae and Cyclobacteraceae* families and *Subgroup 23*, *Actibacter*, *Robiginitalea*, *Spirochaeta 2* and *Sva0081 sediment group* genera (Appendix A). Sites S2 (471) and S6 (426) had the highest number of unique ASVs.

### 3.3. Comparison of Prokaryotic Community Structure

Hierarchical clustering analysis (HCA) based on the Bray-Curtis distance classified samples in two main groups (Figure 4a) and it was significant (*p* ≤ 0.001) as shown by PERMANOVA tests. Group I showed similarities among sites S1 to S4 and Cluster II for samples S5 and S6. Principal coordinate analysis (PCoA) based on the weighted Unifrac distance explained the highest variations for the microbial data in axis 1 (40%) and axis 2 (18.5%) and separates sites S1–S4 from sites S5 and S6 (Figure 4b).

LEFSE analysis for differentially abundant taxa (species and higher taxonomic ranks) for rhizosphere samples revealed LDA scores of more than 2 points (Figure 5). *Deltaproteobacteria* (*Desulfobulbaceae* family and *SEEP-SRB1* genus) were among the taxa with higher LDA scores in sites S5 and S6 along with *Pirellulaceae* family and *Rubripirellula* genus (both belonging to *Planctomycetes* phylum). *Acidobacteria* (*Subgroup 10* and *Subgroup 9*) and *Bacteroidetes* (*Aquibacter*, *Portibacter*, *SJA-28*) phyla were also differentially abundant for S5 and S6.

Site S4 was more abundant for *Sphingomonas* (*Alphaproteobacteria*) and *Adhaeribacter* (*Bacteroidetes*) whereas *Pseudomonas stutzeri*, *Aegiribacteria* phyla and *Ferrovibrio* genus (*Alphaproteobacteria*) were differentially abundant in site S3. Site S2 was differentially abundant for the archaeal phyla *Euryarchaeota* (*Marine Benthic Group D and DHVEG-1* and *Marine Group III* orders within *Thermoplasmata* class) among other bacterial taxa and Site S1 was differential for genus *ADurb.Bin120* (*Anaerolineaceae*).

### 3.4. Effect of Abiotic Factors over Prokaryotic Community Structure

The dbRDA analysis based on Bray-Curtis distances explained 51.7% of total variation on axis 1 and 19.8% on axis 2 (Figure 6a) and indicated the five environmental parameters (PW pH, PW ammonium, PW redox, sand content and organic matter) driving rhizosphere prokaryotic community structure. Collectively, porewater redox and ammonium concentration explained 29% of total variance and separated sites S1–S4 from S5 and S6. Mantel tests using Spearman’s correlations showed significant correlations between this environmental data and the distance matrix (P_FDR_ < 0.01).

Spearman’s correlations (Figure 6b) with fdr correction between taxa and environmental factors showed that abundant classes correlated with organic matter included *Deltaproteobacteria* (r = 0.719, P_FDR_ = 0.033), *Gammaproteobacteria* (r = −0.705, P_FDR_ = 0.033) and *Cloacimonadia* (r = −0.685, P_FDR_ = 0.033). W Salinity was significantly correlated with *Deltaproteobacteria* (r = 0.2539, P_FDR_ = 0.6679), *Planctomycetacia* (r = 0.7053, P_FDR_ = 0.042), *Cloacimonadia* (r = −0.8176, P_FDR_ = 0.004). The only taxa correlated with porewater pH (r = −0.7689, P_FDR_ = 0.02267) and redox (r = 0.7875, P_FDR_ = 0.0123) was *Aegiribacteria Phylum*.

### 3.5. Predicted Functional Profiles

Putative functional profiles using the abundance of taxa with the FAPROTAX database for rhizosphere samples are shown in Figure 7a. The analysis suggested that sites S1 to S4 were associated with nitrite respiration (8.8–20.3%), nitrate respiration (9.1–21.3%) and aerobic chemoheterotrophy (19.7–27.2%). In contrast, nitrogen respiration was less than 1% for sites S5 and S6, and between 7.6 and 19.5% for aerobic chemoheterotrophy. These two sites show a higher percentage for the respiration of sulfur compounds, especially sulfate-respiration (14.4–19.9%), compared to sites S1–S4 (9.1–14.6%). Fermentation (3.8–7.2%) and anaerobic chemoheterotrophy (3.8–6.7%) occurred at similar extents for all sites.

LEFSE analysis on Tax4Fun KEGG functional profiles for all rhizosphere sites showed a variety of predicted functions with an LDA score of more than 3 (Figure 7b). Among other functions, Site S3 was significantly different for environmental adaptation related functional profiles such as two-component system, signal transduction, flagellar assembly, bacterial chemotaxis. Functional profiles related with plant metabolism (metabolism of terpenoids and polyketides and arginine and proline metabolism) and carbon and nitrogen metabolism (amino acid metabolism and starch and sucrose) were also differentially abundant. Site S4 was only associated with Xenobiotics biodegradation and metabolism. Sites S5 and S6 were differentially abundant for genetic information processes, such as translation, replication and repair, folding, sorting and RNA degradation, replication and repair, and energy metabolism, which included carbon fixation pathways in prokaryotes and methane metabolism.

## 4. Discussion

### 4.1. Prokaryotic Core Community

Major classes comprising of the bacterial core community in this study included *Deltaproteobacteria*, *Bacteroidia*, *Gammaproteobacteria* and *Planctomycetia*. These classes have also been reported as most abundant for *T. testudinum*, *S.filiforme*, *Z. marina*, *Z.noltii* and *C. nodosa* rhizospheres in the United States (Florida and California) and Portugal (Culatra Island) and, are recognized as important players in sulfur cycling and organic matter degradation [3,26,27]. In contrast, *Thermoanaerobaculia* (*Acidobacteria*), *Ignavibacteria*, *Latescibacteria* and *Aminicenantia* classes were also found as highly abundant in this study but not for other seagrasses rhizospheres elsewhere. The latter three classes have been recognized for their important roles in nitrogen cycling (*Ignavibacteria and Aminicenantia*) and plant detritus degradation capacities (*Latescibacteria*) [10,52].

At lower taxonomic levels, several bacterial groups involved in sulfur cycling were detected for all the samples. *Desulfobulbaceae* family, *Sva0081 sediment group* genus (*Desulfobacteraceae*), *Spirochaeta 2* and *PHOS-HE36* were among the most abundant. *Desulfobulbaceae* family members have been recognized as both sulfide oxidizing and sulfate-reducing bacteria which act as an important sink of acetate and as key players in carbon cycling in organic rich sediments while *Sva0081 sediment group genus* can also be involved in both sulfate-reduction and sulfide-oxidation (using either oxygen or nitrate) processes and have been detected in rhizospheres of aquatic plants [53,54]. *Spirochaeta 2* is recognized for its sulfur oxidizing capacities in seagrass rhizospheres while *PHOS-HE36* genus can be involved in sulfur oxidation and denitrification processes [7,55]. To the best of our knowledge, this is the first time that *PHOS-HE36* has been reported as a major genus in seagrass rhizosphere studies. Since several major die-offs of seagrasses (including *H.wrightii*) caused by increased sulfide concentrations have been reported around the world, sulfide oxidizing groups in the core community might be vital for seagrass survival under environmental stressors [27,54]. Complex organic matter (rhizodeposits and cell walls polysaccharides) degraders such as *Bacteroidetes BD-2* and *Sandaracinaceae* were also detected within the core community [10,54].

Archaeal community was dominated by *Crenarchaeota*, *Euryarchaeota* and *Asgardaeota* phyla and almost 50% of archaeal sequences were represented by *Bathyarchaeia* class (Appendix A). Similar relative abundances for this class were obtained by [13] in sediments colonized by *Z. japonica* but differ from previous reports suggesting that *Bathyarchaeota* and *Euryarchaeota* are more abundant in mid- and high- latitude aquatic environments [11]. *Bathyarchaeota* recognized by its potential for acetogenesis, dissimilatory nitrite reduction to ammonium, methane production and sulfur cycling [13,56,57], may serve as a keystone species with possible interactions with sulfate-reducing bacteria. Its role in nitrogen (nitrogen fixation, ammonium transporter) and sulfur metabolism (sulfate and thiosulfate reduction) and the capability for degrading detrital proteins, polymeric carbohydrates and fatty acids have also been previously reported [11].

### 4.2. Environmental Drivers and Comparison of Prokaryotic Community Structure

As suggested by hierarchical clustering, sites S1 to S4, where lower ammonium concentrations, lower organic matter contents and coarser textures were recorded, formed a separate group from sites S5 and S6 (Figure 4a). *Pseudomonas stutzeri* was abundant in sites S1 to S4 ranging from 8.8 to 20.3% of relative abundances and undetected in S5 and S6. LEFSE analysis indicated the highest LDA score for this ASV was for site S3 (Figure 5). This bacterium may involve in multiple biogeochemical process, such as nitrate dependent Fe (II) oxidation, thiosulfate oxidation and nitrogen fixation and has already been reported at similar abundances (17.6%) for sediments colonized by *Z. marina* in a temperate region shallow coastal lagoon [58,59]. Since higher inorganic nitrogen concentrations in water samples due to freshwater discharges were recorded, this bacterium could play an important role in removing nitrogen excess by denitrification. Moreover, higher redox potentials and opposite trends of relative abundances between sulfate-reducing and denitrifying taxa were observed as previously found by [60]. This could be explained to the fact that oxygen stimulates while sulfide inhibits nitrification. *Aegiribacteria* phylum was also different for site S3 and is presumed to be involved in fermentative processes [61]. Positive correlations with porewater redox and negative with pH were found for this phylum. *Marine Benthic Group D and DHVEG-1* and *Marine Group III* orders (*Thermoplasmata* class, *Euryarchaeota* phylum) had a high LDA score for site S2, being consistent with previous reports regarding its high sensitivity to temperature and salinity, although no significant correlations with environmental data were found in this study [62]. Acetate and ethanol generation via fermentation and exogenous protein mineralization has been reported for these taxa [11]. Therefore, along *Aegiribacteria*, may play an important role for heterotrophic bacteria.

Conversely, sites S5 and S6 were clustered apart. *Acidobacteria* (*Subgroup 9* class and *Subgroup 10* genus), *Bacteroidetes* (*SJA-28* order, *Aquibacter* and *Portibacter* genus) and *Planctomycetes* (*Rubripirellula genus*) were more abundant for these sites (Figure 5), which have been reported as important players in marine nutrient cycling due to the degradation of complex organic matter [3,63,64]. This is supported by the positive correlations obtained between organic matter and *Planctomycetacia* and the fact that higher organic matter contents were obtained for these sites. *Desulfobulbaceae* family and *SEEP-SRB1* (*Desulfobacteraceae*) were also differentially abundant for these sites. Increased nitrogen concentrations have also been associated with lower abundances of *Deltaproteobacteria* but higher *Gammaproteobacteria* relative abundances for *Thalassia hemprichii* rhizospheres [65] similar to the ones observed in Figure 3 for this study. *SEEP-SRB1* and *Desulfobulbaceae* are acknowledged as complete-oxidizing SRB capable of using a diversity of organic carbon compounds and have been reported as most abundant members within the class *Deltaproteobacteria* in seagrasses from tropical and temperate regions [3,7].

The finer grained textures but increased organic matter contents in sediments and PW-ammonium concentrations recorded in sites S5 and S6 are consistent with the capacity of fine-grained sediments to retain organic matter, ammonium and phosphate reported elsewhere, thereby influencing nutrient availability [66,67]. Since organic matter and PW-ammonium contributed to explain differences in community structure (Figure 6a) and significant correlations with prokaryotic taxa were obtained (Figure 6b), different prokaryotic groups, such as sulfur metabolizing taxa and organic matter degraders could be important for nutrient availability in those SGD uninfluenced sites. In contrast, bacterial groups for nitrogen removal could be important in those sites influenced by SGD.

One major perspective to be accounted for future studies is to include another potentially important environmental parameters driving community structure. This includes dissolved organic carbon in the rhizosphere and surface irradiance. Given that depth increased from 1.3 at the start of the transect to 3 m at the end, and to the fact that roots exudates and oxygen supply are dependent on light requirements, surface irradiance could be an important environmental driver in nutrient-rich gradients. Another future consideration for future studies is the sampling of unvegetated surrounding sediment to assure community structure changes along environmental gradients are due to plant responses and not attributed to the sole effect of changing environmental conditions.

### 4.3. Functional Profiling

As shown by Tax4Fun predicted profiles, site S3 under the influence of SGD, was differentially abundant for amino acid metabolism and environmental adaptation related functional profiles such as signal traduction, flagellar assembly, biofilm formation, two-component system that enable bacteria to sense, respond, and adapt to changes in their environment or in their intracellular state, and bacterial chemotaxis, which is the process by which cells sense chemical gradients in their environment and, then, move towards more favorable conditions [68]. Amino acid metabolism and signal transduction in rhizospheres of tropical seagrass *T. hemprichii* has been found to associate with high levels of inorganic nitrogen, suggesting that seagrasses secrete large amounts of aminoacids promoting this metabolism in bacteria [65,69]. Glycine, serine and threonine metabolism, described as processes helping to maintain the redox balance and energy levels in plants, arginine and proline metabolism (highly beneficial for plants exposed to stress conditions), and metabolism of terpenoids and polyketids, also involved in plant metabolism, were the amino acid metabolism related profiles [46,70]. Moreover, starch and sucrose metabolism, recently acknowledged as the major sugar excreted by *Posidonia oceanica*, was also more expressed for these sites [6]. Among other, sites S5 and S6 were differentially abundant for genetic information processes, vital for proliferation and growth of microorganisms, such as, Translation, Replication and Repair, Folding, sorting and degradation and replication and repair, and Energy metabolism, which included Carbon fixation pathways in prokaryotes and methane metabolism, usually occurring in nutrient-poor environments [71].

Relevant ecological functions mapped using the FAPROTAX database suggested denitrification processes could be more expressed in sites under the influence of SGD and it could be explained by the higher *Pseudomonas stutzeri* abundances obtained for sites S1-S4. Aerobic chemoheterotrophy was also more expressed in these sites and might be supported by the higher redox potentials and possibly increased photosynthate exudation recorded since amino acids and sucrose and starch metabolism were differentially abundant as shown by the TAX4FUN analysis. Conversely, for sites S5 and S6 higher percentages of sulfur compounds respiration are supported by differentially abundant sulfur metabolizing taxa as shown by LEFSE analysis. As suggested, anaerobic chemoheterotrophy and fermentation could be occurring across all sites regardless of the environmental conditions present and could be key processes for seagrass nutrient acquisition and survival mediated by bacterial and archaeal core community taxa.

## 5. Conclusions

The results obtained in this study demonstrated that environmental changes modified the prokaryotic community structure and predicted functional profiles in the rhizosphere of *H. wrightii*. However, a core community was detected despite of the changes of environmental conditions, including archaeal classes such as *Bathyarchaeia* which could be playing an important role for organic matter degradation and possibly interacting with important heterotrophs such as sulfur metabolizing taxa by supplying acetate and other carbon sources. Bacterial taxa such as sulfate-reducers, organic matter degraders, sulfide oxidizers, and nitrogen cycling related bacteria were also part of the core rhizospheric community. Furthermore, results suggested that denitrification might be an important process occurring in rhizospheres of *H. wrightii* influenced by SGD. Thus, all these microbial groups could be considered as important for health and survival of *H. wrightii* in coastal ecosystems.

## Figures and Tables

**Figure 1 microorganisms-11-00494-f001:**
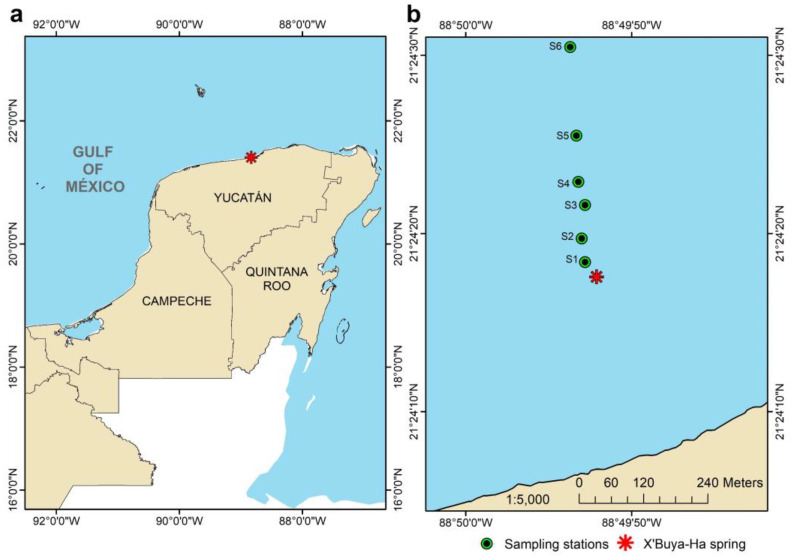
Map showing the Yucatán Peninsula, México, and the location of the study site (red asterisk) (**a**). Transect with the six sampling stations (green and black circles) established along a 400 m gradient in Dzilam de Bravo, Yucatán. X’Buya-Ha spring is shown by a red asterisk (**b**).

**Figure 2 microorganisms-11-00494-f002:**
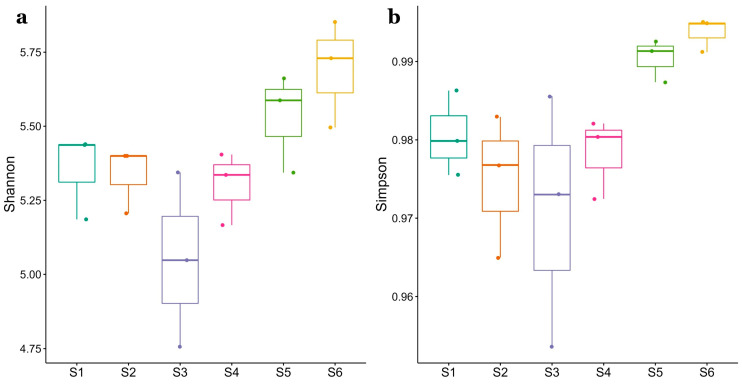
Box plots showing alpha diversity metrics for the six sampling stations (S1–S6). Shannon (**a**) and Simpson (**b**) indexes are presented, and bars are colored by sampling sites. Minimum, maximum, mean values and standard deviation correspond to the triplicate of each of the six sampling stations.

**Figure 3 microorganisms-11-00494-f003:**
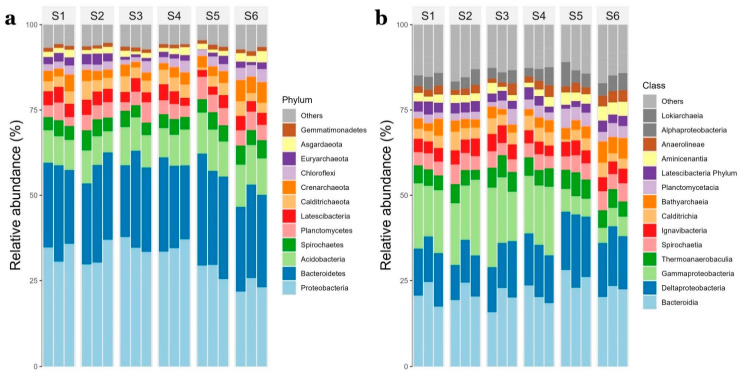
Relative abundance (%) of bacterial and archaeal top phyla (**a**) and classes (**b**). Data are shown for the triplicates of each of the six sampling stations (S1–S6).

**Figure 4 microorganisms-11-00494-f004:**
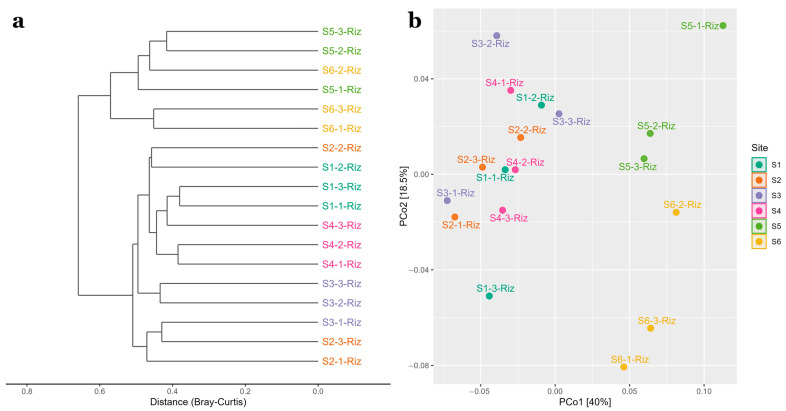
Hierarchical clustering of Bray-Curtis distances for each triplicate of the six sampling stations (S1–S6) (**a**) and PCoA of weighted unifrac distances (**b**) for the same dataset. A color separation by site was applied for both Figures and includes the triplicate data for each site.

**Figure 5 microorganisms-11-00494-f005:**
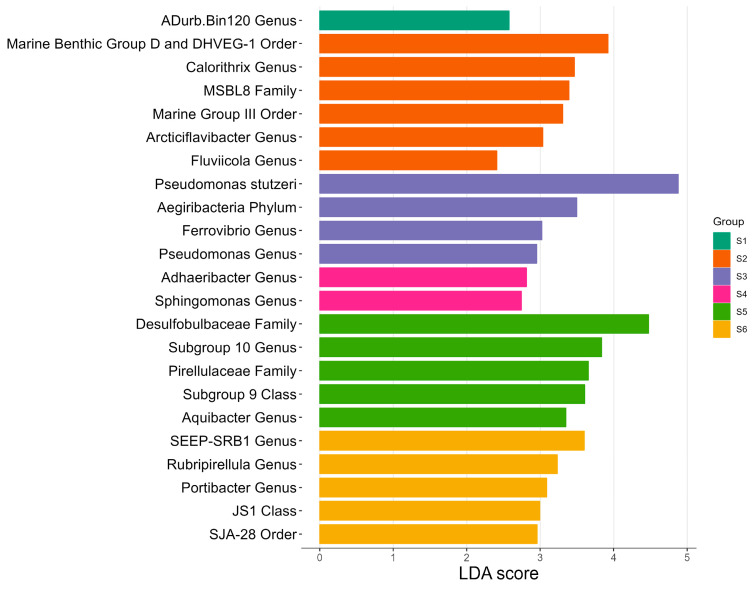
Differential abundances analysis (LEFSE) at species and upper levels for all sampling stations (S1–S6) using triplicate data for each site. An LDA score threshold = 2 was applied.

**Figure 6 microorganisms-11-00494-f006:**
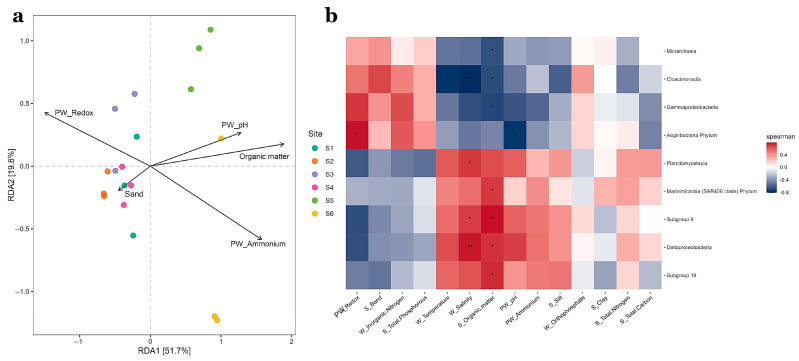
Distance based redundancy analysis (db-RDA) using redox potential, pH and ammonium concentration in porewater, and sand as well as organic matter content for sediments. Total variation percentages are shown for both axis (**a**). A heatmap showing Spearman’s correlations with fdr correction (**b**) between prokaryotic taxa and environmental parameters is also shown. W, PW, and S prefixes denote water, porewater and sediment determinations, respectively. A single asterisk corresponds to significant data with *p*-values less than 0.05 and a double asterisk correspond to *p*-values less than 0.01.

**Figure 7 microorganisms-11-00494-f007:**
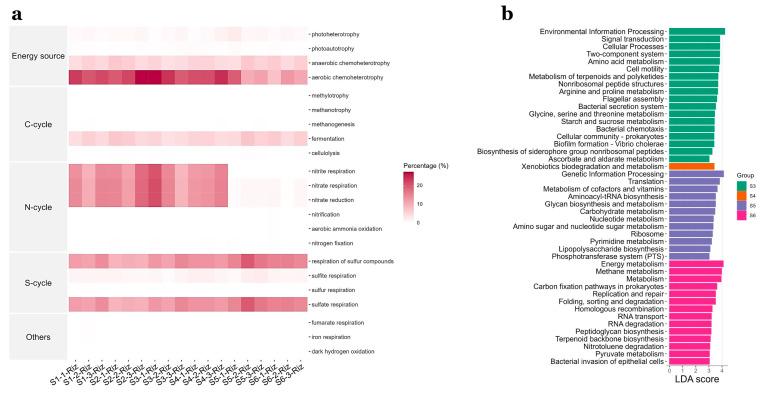
Predicted functional profiles for all sampling sites (S1–S6) using FAPROTAX database using the abundance of taxa. Percentages of taxa with specific traits are shown in the heatmap (**a**) and LEFSE analysis for the same dataset to show the differences across sites. Functional profiles using subsystem level 3 of KEGG Orthologues using Tax4Fun database are shown (**b**).

**Table 1 microorganisms-11-00494-t001:** Environmental characteristics and nutrient concentrations for water and porewater for all sampling stations. Inorganic nitrogen concentration (N_inorg_) included ammonium, nitrate and nitrate contributions. Average of triplicates and standard deviation are presented.

	Water				Porewater		
**Site**	**Salinity (PSU)**	**Temperature (°C)**	**N_inorg_ (µM)**	**Orthophosphate (µM)**	**pH**	**ORP (mV)**	**Ammonium (µM)**
S1	30.83 ± 0.13	26.32 ± 1.02	14.26 ± 0.47	0.5 ± 0.033	7.07 ± 0.20	−54.03 ± 7.82	17.23 ± 0.61
S2	29.2 ± 1.05	25.7 ± 0.94	7.02 ± 0.1	0.2 ± 0.031	7.17 ± 0.06	−56.67 ± 2.5	11.92 ± 0.60
S3	31.4 ± 0.40	27.23 ± 0.45	9.29 ± 0.11	0.22 ± 0.015	7.13 ± 0.07	−55.13 ± 3.52	10.52 ± 1.27
S4	33.57 ± 0.03	28.34 ± 0.04	11.19 ± 0.53	0.17 ± 0.17	7.14 ± 0.09	−55.6 ± 4.75	12.71 ± 2.16
S5	33.62 ± 0.34	27.93 ± 0.34	4.86 ± 0.44	0.26 ± 0.04	7.41 ± 0.04	−69.53 ± 1.66	15.48 ± 0.85
S6	34.46 ± 0.18	29.02 ± 0.02	2.63 ± 0.24	0.08 ± 0.01	7.34 ± 0.09	−65.4 ± 6.09	27.5 ± 0.71

N_inorg,_ NH_4_^+^ + NO_3_^−^ + NO_2_^−^; ORP, Oxidation-Reduction Potential.

**Table 2 microorganisms-11-00494-t002:** Physicochemical characteristics, organic matter content (O.M%) and elemental analysis (nitrogen and carbon) for sediments of sampling sites. Average values and standard deviations were used.

	Sediment						
**Site**	**O.M (%)**	**Sand (%)**	**Silt (%)**	**Clay (%)**	**TP (µmol g^−1^)**	**TN (%)**	**TC (%)**
S1	1.75 ± 0.21	93.42 ± 1.2	3.95 ± 1.26	2.63 ± 0.08	7.27 ± 0.23	0.36 ± 0.075	12.03 ± 1.35
S2	1.34 ± 0.15	94.93 ± 0.06	1.12 ± 0.11	3.95 ± 1.17	6.56 ± 0.92	0.28 ± 0.05	13.44 ± 1.71
S3	1.84 ± 0.1	94.33 ± 1.03	2.15 ± 0.18	3.52 ± 1.15	6.77 ± 0.59	1.57 ± 0.37	12.15 ± 1.43
S4	1.75 ± 0.09	89.1 ± 0.82	7.03 ± 0.85	3.87 ± ±0.03	6.88 ± 0.45	1.38 ± 0.39	12.56 ± 2.31
S5	2.27 ± 0.06	92.15 ± 1.98	4.58 ± 0.31	3.27 ± 1.18	6.78 ± 0.12	1.03 ± 0.18	13.31 ± 1.77
S6	2.33 ± 0.1	87.83 ± 1.66	8.48 ± 2.03	3.68 ± 1.42	6.41 ± 0.11	1.12 ± 0.12	14.16 ± 1.68

O.M%, organic matter; TP, total phosphorous; TN, total nitrogen; TC, total carbon.

## Data Availability

Raw high-throughput sequencing reads were deposited in the National Center for Biotechnology Information (NCBI) Sequence Read Archive (SRA) GenBank database (BioProject ID number: PRJNA927344).

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
