# Peer review of "Changes in the Rhizosphere Prokaryotic Community Structure of Halodule wrightii Monospecific Stands Associated to Submarine Groundwater Discharges in a Karstic Costal Area"

_microorganisms, 2023, doi:10.3390/microorganisms11020494_

Round 1
Reviewer 1 Report
As general comment the work is well written and designed with relevant results.
The authors touch upon very important issues about the changes in the rhizosphere prokaryotic community structure of Halodule wrightii.
This manuscript timely and I commend the authors for bringing in some new ideas and analysis.
This study is very interesting and conforms to the requirements of the Microorganisms journal.
The manuscript is well written.
Materials and method section is well described and correspond to the aim set out in the manuscript.
The results are correctly described.
Figures and tables are clear and understandable.
The discussion is correct.
The conclusions of the article are correct and consistent with the discussed issues.
The references are properly chosen and cited.
I recommend the publication of this manuscript in the Microorganisms journal in present form.
Author Response
Thank you very much for the positive comments.
Reviewer 2 Report
This research topic seems exciting and appropriate for publication in Microorganisms (ISSN 2076-2607). The manuscript, titled "Changes in the rhizosphere prokaryotic community structure of Halodule wrightii monospecific stands associated to submarine groundwater discharges in a karstic costal area" (microorganisms-2210617), explores how environmental stressors impact upon interactions between seagrasses and their prokaryotic community in coastal ecosystems. Content information appears to be fine overall. Prior to submitting the manuscript for fresh review, several significant shortcomings must be addressed.
Following are the details of the comments.
1. The abbreviation must be explained in full rather than starting with the first use throughout the manuscript. See H. wrightii in abstract.
2. The introduction is somehow short and lacks sufficient information. In their detailed descriptions, authors may consider research gaps, their significance, and innovation aspects.
3. Make sure to use the latest information and avoid using old references like. In most cases, authors relied on old literature, which is not preferred by reviewers.
4. The aims should be elaborated more clearly by splitting them into points that correspond to the headings of results and discussion.
5. Site description was not well described. Add more information about the site description.
6. Figure 1 does not meet the required quality, possibly due to the low resolution of the pixels. The font size is too small to be readable. The caption lacks details that are shown in the figures. Revise the figure in response to comments made.
7. It is confusing that the authors used correlation heatmaps (Pearson and Spearman), since data can be parametric or non-parametric. In statistical analysis, this should be explained clearly.
8. The abbreviations used in Tables 1 and 2 must be explained in the notes below these tables so that the reader may have a better understanding of the abbreviations.
9. Figure 2 does not meet the required quality, possibly due to the low resolution of the pixels. The font size is too small to be readable. The caption lacks details that are shown in the figures. Revise the figure in response to comments made.
10. Figure 3 does not meet the required quality, possibly due to the low resolution of the pixels. The font size is too small to be readable. The caption lacks details that are shown in the figures. Revise the figure in response to comments made.
11. Figure 4 does not meet the required quality, possibly due to the low resolution of the pixels. The font size is too small to be readable. The caption lacks details that are shown in the figures. Revise the figure in response to comments made.
12. Figure 5 does not meet the required quality, possibly due to the low resolution of the pixels. The font size is too small to be readable. The caption lacks details that are shown in the figures. Revise the figure in response to comments made.
13. Figure 6 does not meet the required quality, possibly due to the low resolution of the pixels. The font size is too small to be readable. The caption lacks details that are shown in the figures. Revise the figure in response to comments made.
14. Figure 7 does not meet the required quality, possibly due to the low resolution of the pixels. The font size is too small to be readable. The caption lacks details that are shown in the figures. Revise the figure in response to comments made.
15. There is no information regarding the limitations of the study.
16. There is room for improvement in the quality of the language.
Author Response
- The abbreviation must be explained in full rather than starting with the first use throughout the manuscript. See H. wrightii in abstract
Reply: Revision addressed as requested. Changes can be verified in line 11.
- The introduction is somehow short and lacks sufficient information. In their detailed descriptions, authors may consider research gaps, their significance, and innovation aspects.
Reply: Introduction was restructured. Specific information was included to highlight research gaps, their significance and how this study addressed these issues. All this information is highlighted in purple.
- Make sure to use the latest information and avoid using old references like. In most cases, authors relied on old literature, which is not preferred by reviewers.
Reply: Old references were replaced by recent literature. All these replacements were highlighted in cyan color throughout manuscript. i) (Fourqurean et al., 1992; Hemminga & Duarte, 2000) were replaced by (Reynolds et al., 2016; Nordlund et al., 2016) in line 31, ii) (Lee & Dunton, 1999) was replaced by (Jiang et al., 2022) in line 60, iii) (Fourqurean et al., 1992) was replaced by (Darnell et al., 2021) in line 73, iv) (Bulthuis et al., 1992) was replaced by (Krause et al., 2022; Krause-Jensen et al., 2011) in line 421, (Kanehisa & Goto, 2000) was replaced by (Karmakar, 2021) in line 445, v) (Hayat et al., 2012) was excluded from line 452.
- The aims should be elaborated more clearly by splitting them into points that correspond to the headings of results and discussion.
Reply: Specific goals were listed in lines 81-86.
- Site description was not well described. Add more information about the site description.
Reply: The requested information was included in Lines 91-95 and 99-103.
- Figure 1 does not meet the required quality, possibly due to the low resolution of the pixels. The font size is too small to be readable. The caption lacks details that are shown in the figures. Revise the figure in response to comments made.
Reply: The resolution of Figure 1 is now 450 dpi in the Word document and the compressed file. Font size was increased, and every depicted detail is now described in the caption.
- It is confusing that the authors used correlation heatmaps (Pearson and Spearman), since data can be parametric or non-parametric. In statistical analysis, this should be explained clearly.
Reply: Pearson's correlations in the results section (former Figure 6c) and every paragraph connected to this analysis, were removed to avoid confusing the readers. We decided this because Spearman correlations at Class level were sufficient to support our findings.
- The abbreviations used in Tables 1 and 2 must be explained in the notes below these tables so that the reader may have a better understanding of the abbreviations.
Reply: Revision addressed as requested for Table 1 and Table 2.
- Figure 2 does not meet the required quality, possibly due to the low resolution of the pixels. The font size is too small to be readable. The caption lacks details that are shown in the figures. Revise the figure in response to comments made.
Reply: Image was exported with a resolution of 330 dpi in both the document and compressed file. Image settings and resizing were applied for a better display. Additional information to the Figure caption was included.
- Figure 3 does not meet the required quality, possibly due to the low resolution of the pixels. The font size is too small to be readable. The caption lacks details that are shown in the figures. Revise the figure in response to comments made.
Reply: Image was exported with a resolution of 330 dpi in both the document and compressed file. Image settings and resizing were applied for a better display. Font size was increased.
- Figure 4 does not meet the required quality, possibly due to the low resolution of the pixels. The font size is too small to be readable. The caption lacks details that are shown in the figures. Revise the figure in response to comments made.
Reply: Image was exported with a resolution of 330 dpi in both the document and compressed file. Image settings and resizing were applied for a better display. Additional information to the Figure caption was included.
- Figure 5 does not meet the required quality, possibly due to the low resolution of the pixels. The font size is too small to be readable. The caption lacks details that are shown in the figures. Revise the figure in response to comments made.
Reply: Image was exported with a resolution of 400 dpi in both the document and compressed file. Image settings and resizing were applied for a better display. Font size was increased. Additional information to the Figure caption was included.
- Figure 6 does not meet the required quality, possibly due to the low resolution of the pixels. The font size is too small to be readable. The caption lacks details that are shown in the figures. Revise the figure in response to comments made.
Reply: Image was exported with a resolution of 330 dpi in both the document and compressed file. Image settings and resizing were applied for a better display. Font size was increased. Additional information to the Figure caption was included.
- Figure 7 does not meet the required quality, possibly due to the low resolution of the pixels. The font size is too small to be readable. The caption lacks details that are shown in the figures. Revise the figure in response to comments made.
Reply: Image was exported with a resolution of 330 dpi in both the document and compressed file. Image settings and resizing were applied for a better display. Font size was increased. Additional information to the Figure caption was included.
- There is no information regarding the limitations of the study.
Reply: Major perspectives from this study regarding unrecorded potential environmental drivers and control sediment (unvegetated surrounding sediments) were described in lines 428-436
- There is room for improvement in the quality of the language.
Reply: Grammar improvements through the manuscript were applied.

Round 2
Reviewer 2 Report
The revised draft has been substantially revised by the authors. Nevertheless, I have a few minor concerns. The first is that the authors did not properly maintain the track changes option, so it is not easy to see all the changes they have made.
Secondly, they need to adjust the numbering of their objectives.
Thirdly, the references are not properly presented as advised by the journal.
Author Response
The following changes has been applied. I) The Track Changes tool was applied to the manuscript submitted during round 1, and now every modification to the original manuscript can be tracked, including the changes requested on 09 Feb 2023 (this version of the manuscript ist attached in non published material), II) The numbering of the objectives described at the end of the introduction section has been corrected, and III) The reference list at the end of the document was modified to comply with the requirements of the journal’s criteria
The previous version of the manuscript with track changes tool can be found in non-published material section.